



# Satellite observations of aerosols and clouds over South China from 2006 to 2015: analysis of changes and possible interactions

Nikos Benas[1], Jan Fokke Meirink[1], Karl-Göran Karlsson[2], Martin Stengel[3], Piet Stammes[1]

[1]Royal Netherlands Meteorological Institute (KNMI), De Bilt, the Netherlands

[2]Swedish Meteorological and Hydrological Institute (SMHI), Norrköping, Sweden

[3]Deutscher Wetterdienst (DWD), Offenbach, Germany

*Correspondence to*: Nikos Benas (benas@knmi.nl)

**Abstract.** Aerosol and cloud properties over South China during the 10-year period 2006-2015 are analysed based on observations from passive and active satellite sensors and emission data. The results show a decrease in aerosol optical depth

over the study area by about 20% on average, accompanied by an increase in liquid cloud cover and cloud liquid water path (LWP) by 5% and 13%, respectively. Analysis of aerosol types and emissions suggests that the main driver for their reduction is a decrease in biomass burning aerosols. These changes occurred mainly in late autumn and early winter months and coincided with changes in cloud properties. For the latter, possible explanatory mechanisms were examined, including changes in circulation patterns and aerosol-cloud interactions. Further analysis of changes in aerosol vertical profiles

demonstrates a consistency of the observed aerosol and cloud changes with the aerosol semi-direct effect, which depends on their relative heights. Based on this mechanism, less absorbing aerosols in the cloud layer would lead to an overall decrease in evaporation of cloud droplets, thus increasing cloud LWP and cover.

## 1 Introduction

The role of atmospheric aerosols in climate change has been studied widely in the past. Their various effects are broadly

defined based on their interactions with atmospheric radiation and clouds. The direct effect is described through scattering and absorption of radiation whereas indirect effects describe interactions with clouds, which can lead to changes in both cloud albedo (Twomey, 1977) and cloud lifetime (Albrecht, 1989). The semi-direct effect is a third category that describes aerosol-induced changes in clouds through interaction with radiation. According to the latest terminology (Boucher et al., 2013), the semi-direct effect is described as a "rapid adjustment" induced by aerosol radiative effects, and along with the

direct effect it is grouped into the "Aerosol-Radiation Interactions" (ARI) category, whereas the indirect effects are termed "Aerosol-Cloud Interactions" (ACI).

Observations of these mechanisms and their effects on climate have been elusive, and the uncertainties associated with them remain high (Boucher et al., 2013). The main reasons for this lack of substantial progress originate in the high complexity of these phenomena, with multiple possible feedback mechanisms and dependences on various parameters in different regimes

(Stevens and Feingold, 2009, Bony et al., 2015). Although there are continuous improvements, the mechanisms related to aerosol and cloud interactions and feedbacks are still inadequately represented in models (Feingold et al., 2016), and poorly captured by remote sensing measurements (Seinfeld et al., 2016). Regarding the latter approach, many studies have highlighted the difficulties and limitations of remote sensing methods, which usually include limitations in spatial and temporal samplings (Grandey & Stier, 2010; McComiskey & Feingold, 2012). On the other hand, progress is steadily being

made, as data sets of aerosols and clouds based on remote sensing retrievals gradually improve. Additionally, independent data sets with complementary characteristics and properties become constantly available, allowing more in-depth analyses of



the aerosol and cloud conditions and opening new possibilities for synergistic usage, towards further constraining the effects of aerosols on clouds.

The present study builds on these developments by providing an analysis of aerosol and cloud characteristics and changes in recent years over a climatically important and sensitive area in South China. This region (20°-25° N, 105°-115° E) was
selected, being a densely populated area with intense human activities, ranging from urban and industrial to agricultural, which also constitute different sources of aerosol emissions. Furthermore, significant changes in aerosol loads during the past years over the wider surroundings have previously been reported (e.g. Zhao et al., 2017; Sogacheva et al., 2018), providing the opportunity for an analysis of possible effects on clouds. Hence, the purpose of this study is dual. The primary aim is to analyse aerosol and cloud characteristics and changes during the previous years over South China. Using multiple
data sets, created based on different retrieval approaches, adds robustness to the results. The secondary purpose of this study is to investigate the possibilities and limitations of the synergistic use of this multitude of aerosol and cloud data sets for the assessment of possible aerosol and cloud interactions. For this purpose, data sets are analysed in combination, to either help exclude possible explanatory mechanisms, or provide indications of their manifestation.

The study is structured as follows: Section 2 provides a description of the aerosol and cloud data sets used, and the
methodology for analysing their changes. Results of this analysis are described in Section 3, including time series and seasonal changes in aerosols and clouds, possible effects of large-scale meteorological variability, and indications of possible effects of aerosol changes on corresponding cloud changes. Our findings are summarized in Section 4.

## 2 Data and methodology

### 2.1 Aerosol data

Analysis of aerosol changes was based on MODerate resolution Imaging Spectroradiometer (MODIS) and Cloud-Aerosol Lidar and Infrared Pathfinder Satellite Observations (CALIPSO) data. MODIS is a sensor on board NASA's Terra and Aqua polar orbiters, providing aerosol and cloud data products since 2000 and 2002 from Terra and Aqua, respectively. The Aqua MODIS level 3 Collection 6 daily Aerosol Optical Depth (AOD) was used here, available over both land and ocean at $1° \times 1°$ spatial resolution (Levy et al., 2013).

The CALIPSO level 3 monthly aerosol profile product was used along with MODIS, to include information on different aerosol types and their vertical distribution in the analysis. CALIPSO level 3 parameters were derived from the corresponding instantaneous level 2 version 3 aerosol product (Winker et al., 2009; Omar et al., 2009) and include column AOD of total aerosol, dust, smoke and polluted dust, available globally at $2° \times 5°$ latitude/longitude resolution, along with their extinction profiles at 60 m vertical resolution, up to 12 km altitude.

Apart from the characterization of aerosol loads and vertical distributions over the region with MODIS and CALIPSO data, aerosol sources were investigated using the Global Fire Emissions Database (GFED), which provides information about trace gas and aerosol emissions from different fire sources on a global scale. Here, version 4 of the data set was used (GFED4s), available at $0.25° \times 0.25°$ spatial resolution and on a monthly basis. GFED emission estimates are based on data of burned areas and active fires, land cover characteristics and plant productivity, and the use of a global biogeochemical
model (Van der Werf et al., 2017). It should be noted that, due to the long-range transport of aerosols, local aerosol emissions are not expected to fully explain corresponding properties and characteristics of aerosol types and loads in the atmosphere of the same region. Emission data were rather used here for partially explaining the origin of aerosol types and distributions detected from space. They were also useful as an indicator of local aerosol-producing human activities, with biomass burning being a major source.



## 2.2 Cloud data

Two independently derived, satellite-based cloud data sets, were used for the analysis of cloud properties and changes over South China. The Aqua MODIS level 3 Collection 6 daily $1° × 1°$ product was used (Platnick et al., 2017), as in the case of AOD, for the estimation of monthly averages and corresponding changes in cloud properties, including total and liquid

Cloud Fractional Coverage (CFC), in-cloud and all-sky Liquid Water Path (LWP), as well as liquid Cloud Optical Thickness (COT) and Effective Radius (REFF).

The same cloud properties were analyzed using the second edition of the Satellite Application Facility on Climate Monitoring (CM SAF) cLoud, Albedo and surface RAdiation data set from AVHRR data (CLARA-A2), a recently released cloud property data record, created based on Advanced Very High Resolution Radiometer (AVHRR) measurements from

NOAA and MetOp satellites (Karlsson et al., 2017). It covers the period from 1982 to 2015 and includes, among other parameters, CFC and cloud phase (liquid/ice), cloud top properties and cloud optical properties, namely COT, REFF and water path, separately for liquid and ice clouds. CLARA-A2 level 3 data, available at $0.25° × 0.25°$ spatial resolution, from the afternoon satellites NOAA-18 and NOAA-19, were analyzed in the present study.

The use of two independent cloud data sets, derived from different sensors, increases confidence in the results on cloud

changes and seasonal variability. The similarity between these results strongly suggests that the detected changes can be attributed to clouds themselves, instead of possible sensor degradations or retrieval artifacts. The same holds for the aerosol analysis, for which three independent data sets, with different retrieval approaches, were used.

## 2.3 Analysis of changes

The quantification of changes during the study period was based on linear regression fits to the deseasonalized monthly time

series. Initial monthly values were computed as the area-weighted spatial averages of the study region. Deseasonalization was then performed by subtracting from each month the corresponding time series average of this month and then adding the average of all months in the time series. For every aerosol and cloud variable $X$ studied, the change $\Delta X$ was calculated as $\Delta X = X_f – X_i$, where $X_i$ and $X_f$ are the initial and final monthly values of the regression line. The corresponding percent change was estimated as $\Delta X = 100(X_f – X_i)/ X_i$. Spatial and temporal representativeness of the study area and time period in the

change analysis were ensured by applying thresholds to both the area covered with valid data and the number of months used in the calculations. Specifically, the following thresholds were applied: a) on a pixel basis, a monthly average value was used only if it was computed from at least 18 daily values (10 daily values for AOD, due to sparsity of data); b) a spatially averaged monthly average value was used if it was computed from at least 50% of the pixels in the study area; c) it was required that at least 80% of monthly averages are present in the time series, for the corresponding 10-year changes to be

estimated. Further analysis included a per month estimation of changes, in order to assess their seasonal variation. In this case, no deseasonalization was applied. Statistical significance of all calculated changes was estimated using the two-sided t-test.

## 3 Results

### 3.1 Aerosol characteristics and changes

Aerosol sources in South China include biomass burning activities, such as residential biofuel consumption, crop residues burning, firewood consumption and agricultural waste open burnings (Chen et al., 2017). These sources exhibit different seasonal characteristics and relative contributions to the total aerosol load. Higher emissions of domestic biomass burning occur in autumn and winter, specifically November to March (He et al., 2011), while agricultural field fires are mostly observed after harvesting seasons, when rice and wheat straw field burning takes place, typically in late May and October



(Zha, 2013; Chen et al., 2017). Domestic burning is the major contributor, reaching over 60% of the total biomass burning emissions (He et al., 2011).

Figure 1 shows the seasonal variation of emissions from GFED and AOD from MODIS and CALIPSO over South China, based on data during 2006-2015. The seasonal variation of carbon emitted from biomass burning over the region shows that the highest emissions occur between November and April (Fig. 1a). This seasonal pattern in biomass burning carbon emissions is in good agreement with the seasonal variation of biomass burning activities described before, verifying the high contribution of domestic fuelwood burning during the same months. MODIS and CALIPSO total AOD (Fig. 1b) are in relatively good agreement in most months, with the largest differences occurring in March and April. Some disagreement between MODIS and CALIPSO should be expected, considering their differences in areas sampled, overpass times and retrieval methodologies. Based on CALIPSO, which offers additional information on aerosol types, smoke and polluted dust have similar AODs, while the contribution of dust is minimal, with a small peak in spring. According to the CALIPSO classification, smoke aerosols originate in biomass burning activities, and polluted dust aerosols are a mixture of dust with biomass burning smoke (Omar et al., 2009). Biomass burning emissions and satellite-based AOD are not directly comparable. Apart from additional aerosol sources that contribute to the total AOD and are not represented in GFED, transportation of aerosols from neighbouring regions can also cause large differences. In fact, high smoke AOD values combined with low emissions (e.g. in September and October), suggest that these aerosols were transported to the study region from different areas.

Figure 2 shows the changes in AOD over the South China region during the 10-year period examined, both on a pixel basis (Fig. 2a) and as spatially averaged time series (Figs. 2b and 2c). The pixel-based changes in AOD (Fig. 2a), deduced from daily MODIS level 3 data, reveal an almost uniform reduction throughout the area, with stronger decreases over land. The time series of the deseasonalized spatially averaged monthly values of the AOD, separately from MODIS and CALIPSO, are shown in Figs. 2b and 2c, along with their linear regression fits and corresponding changes (in percent). The reduction in total AOD during the 10-year period is apparent and statistically significant in the 95% confidence interval in both MODIS and CALIPSO data. Based on the CALIPSO aerosol types classification, this decrease can be attributed to corresponding reductions in polluted dust and smoke aerosols. Since no significant change is found for pure dust aerosols, the decrease in polluted dust AOD can also be attributed to biomass burning aerosols. The reduction in AOD reported here is in agreement with changes over the same region or wider Chinese regions during recent years, reported based on different satellite sensors, e.g. MODIS and AATSR (Sogacheva et al., 2018) and MODIS and MISR (Zhao et al., 2017).

The seasonality variability of aerosols over the study region (Fig. 1) suggests that their changes could also exhibit seasonal variations. Hence, the time series changes were further analyzed in terms of their seasonal variability. Results for both AOD and emissions are shown in Fig. 3. For AOD, the main decrease occurs in autumn and early winter. MODIS (Fig. 3a) and CALIPSO (Fig. 3b) agree well in this seasonal pattern. Based on CALIPSO, this decrease is driven by biomass burning aerosols: as for the full time series (Fig. 2c), dust aerosols show no significant change, hence the changes in polluted dust should also be attributed to reductions in biomass burning aerosols. The same analysis of the total mass of carbon particles (C) from local emissions (Fig. 3c) shows that the largest decrease in emitted particles occurs during late autumn to early spring, with a minimum in November, suggesting that this decrease should be attributed to changes in residential energy sources, which peak during the same period. This explanation is also consistent with previous studies, which report a diminishing contribution of residential biomass burning, starting already in the 1990s (Qin and Xie, 2011; 2012; Streets et al., 2008), mainly through a replacement of fuelwood by electricity (Yevich and Logan, 2003). Furthermore, a direct comparison of changes in satellite-based AOD and surface emissions offers additional insights into the origins of these changes: the seasonal variation of changes in C emissions partially agrees with the total AOD change pattern, while this agreement improves in the case of polluted dust. These results suggest that part of the aerosol load over the study area (especially smoke aerosols) is transported from neighboring regions, as was also inferred from the differences in seasonality





patterns (Fig. 1). In such cases, AOD and local emissions do not agree well (e.g. smoke aerosols in October). Forest fires and biomass burning activities in Indochina could be such sources. In November, on the other hand, AOD probably originates mainly from local sources, leading to a coincidence in AOD and emission reductions.

### 3.2 Cloud characteristics and changes

The seasonality of main cloud properties over the study region, comprising total and liquid cloud cover, and optical thickness and effective radius for liquid clouds, is shown in Fig. 4. While the total cloud cover does not exhibit strong seasonal characteristics (Fig. 4a), varying between 0.7 and 0.8 throughout the year (based on CLARA-A2 and MODIS, respectively), liquid clouds appear to prevail from late autumn to early spring (Fig. 4b). A similar seasonal pattern appears in liquid COT, which is not necessarily related to the variation in the extent of liquid clouds. Liquid REFF ranges between 10 μm and 14 μm

throughout the year. The LWP, which proportional to the product of liquid COT and REFF, also varies seasonally, with higher values in winter (not shown here). The main driving factor for the seasonality in total and liquid cloud cover is the Asian Monsoon (AM), which leads to more clouds in summer compared to winter, but more liquid clouds in winter (Pan et al., 2015). The prevalence of low, liquid clouds in winter, which are mostly single-layer clouds, is also verified based on CALIPSO data (Cai et al., 2017).

Figure 5 shows pixel based and spatially averaged changes in cloud properties over South China during the period examined. The all-sky LWP and liquid CFC have increased over most parts of the land and significantly in most cases (Figs. 5a and 5b). In fact, Fig. 5 shows increases in all liquid cloud properties, with the largest increase found for the total liquid water content present in clouds (12%-14%). Liquid COT changes appear similar to those of LWP, with very good agreement between the two data sets (CLARA-A2 and MODIS), while liquid REFF changes are also positive but more ambiguous.

Cloud changes appear statistically significant at the 95% level over large areas of the study region, especially over land, when studied on a pixel basis. Analysis of spatially averaged values, however, over the entire (5° × 10°) study region, reduces this significance to levels below 95% in most cases of Fig. 5. Overall, MODIS and CLARA-A2 are in good agreement and consistent in terms of the changes reported, with biases of around 10% appearing for liquid CFC (Fig. 5d) and REFF (Fig. 5f).

The long time range available from CLARA-A2 data (34 years, starting in 1982) offers the opportunity for further evaluation of the cloud properties changes reported before, especially with respect to changes during the past three decades. For this purpose, changes from all possible time ranges, at least 10 years long and starting from 1982 onward, were estimated for the study region. Results, shown in Fig. 6, suggest that the ranges of changes reported in Fig. 5 are not typical of the entire 34-year CLARA-A2 period. Specifically, for LWP, liquid CFC and liquid COT, the largest increases occur when the time range examined

ends within the last five years of the CLARA-A2 period (2011-2015), indicating that corresponding values reached maxima during these years. Furthermore, for liquid REFF, a switch in the sign of change appears in the last years: while liquid REFF is mainly decreasing for most start and end year combinations, only positive changes appear after 2003, indicating a consistent increase during the last years. It should be noted that abrupt changes appearing in the plots of Fig. 6 should be attributed to artifacts especially in the early years of the CLARA-A2 data record. Specifically, negative changes in

liquid CFC occurring for starting years between 1988 and 1994 coincide with the period when AVHRR on NOAA-11 was operational, which caused a small discontinuity in the time series. Additionally, the switch from channel 3b (at 3.7 μm) to channel 3a (at 1.6 μm) on NOAA-16 AVHRR during 2001-2003 caused a discontinuity in the cloud property time series, most prominently visible for REFF.

As for aerosols, the seasonality of cloud property changes was also analyzed. Figure 7 shows that the overall increase in

liquid clouds during the 10-year period examined can be attributed to changes occurring mainly in November and December. In fact, the patterns of seasonal changes show that CLARA-A2 and MODIS agree very well, with an increase in LWP occurring primarily in December and secondarily in November (Fig. 7a), and liquid CFC increases prevailing also in





November and December (Fig. 7b). Corresponding results for liquid COT and liquid REFF (Figs. 7c and 7d) indicate the similarity in change patterns between COT and LWP, and the ambiguity in the REFF change between CLARA-A2 and MODIS, especially in November. The liquid CFC change is statistically significant in the November case, while all other cloud property changes shown in Fig. 7 are significant in December.

## 5 3.3 Discussion

### 3.3.1 Possible effects of meteorological variability and large-scale phenomena

The results presented in the previous section show that during the study period, aerosols decreased over South China particularly in autumn and early winter, while liquid clouds increased mainly in late autumn and early winter. Hence, there is a concurrence of substantial aerosol and cloud changes during the same months, namely in late autumn and early winter.

There are two major mechanisms that could lead to this concurrence: large-scale meteorological variability could affect both aerosols and clouds simultaneously, while local-scale ACI and/or ARI mechanisms, would lead to cloud changes due to corresponding aerosol changes. A combination of these two factors should also not be excluded.

In order to analyse meteorological variability, namely changes in atmospheric circulation patterns and their possible role in the changes reported before, we used surface pressure and 500 hPa geopotential height fields from the Copernicus

Atmospheric Monitoring Service (CAMS) reanalysis data record (Flemming et al., 2015; 2017). Similarly to the aerosol and cloud properties, the analysis was based on deseasonalized linear regressions of the entire time series of monthly averages, as well as changes on a monthly basis, focusing especially on months when aerosol and cloud changes maximize (i.e. November-December). For this analysis, however, the study area was extended by 10° in every direction, to include large-scale patterns that could be affecting the South China region.

The analysis showed 500 hPa geopotential height changes at the pixel level in the order of several meters and surface pressure changes up to a few hPa, none of which were statistically significant, when either the entire time series or specific months were examined. These results suggest that meteorological variability is not among the major factors contributing to the aerosol and cloud changes reported.

Changes in atmospheric circulation could also be related to larger scale phenomena affecting the wider South-East Asia

region, namely the El Nino Southern Oscillation (ENSO) and Asian Monsoon (AM) cycles. Regarding possible effects of ENSO over South China, the Oceanic Nino Index (ONI) was used to examine possible correlations between ENSO and the aerosol and cloud properties analysed here. ONI is the National Oceanic and Atmospheric Administration (NOAA) primary indicator for measuring ENSO; it is defined as the 3-month running Sea Surface Temperature (SST) anomaly in the Nino 3.4 region, based on a set of improved homogeneous SST analyses (Huang et al., 2017). This analysis showed no particular

correlation between ONI and cloud or aerosol properties; Correlation coefficients were around -0.2 for the entire time series and slightly larger for specific months. A very similar, not significant, anti-correlation between ENSO and low cloud amount was found by Liu et al. (2016), examining the entire China and the period 1951-2014.

The overall effects of AM on the area are most pronounced in summer. Although AM is known to affect aerosol concentrations (through wet deposition during the raining season) and cloud cover, this seasonality pattern does not coincide

temporally with the seasonal aerosol and cloud changes reported here. Furthermore, it is known that AM and ENSO are strongly correlated (Li et al. 2016), hence the effects of the former on these changes are expected to be similarly insignificant with those of the latter.

### 3.3.2 Possible effects of ACIs and ARIs

Although cause and effect mechanisms cannot be proven based on observations only, possible underlying ACI and ARI

mechanisms are worth investigating, since the combination of aerosol and cloud changes can also be used to exclude some of them.



Following this approach, our results appear inconsistent with the standard definitions of the first and second aerosol indirect effects, although the possibility of multiple mechanisms occurring simultaneously cannot be excluded. Specifically, according to the first aerosol indirect effect, a decrease in aerosols would lead to an increase in cloud droplet size, under constant liquid water content. In our case, while both CLARA-A2 and MODIS indicate an overall increase in liquid REFF

(Fig. 5f), these changes do not coincide seasonally with any significant aerosol change, which occurred mainly in autumn (Fig. 3). In fact, mixed signs in liquid REFF change were observed in November (Fig. 7d). Additionally, the LWP increases considerably, suggesting that the first indirect effect mechanism does not play a major role. According to the second aerosol indirect effect, a decrease in aerosols implies reduced cloud life time through more precipitation. However, the increase in observed cloud fraction suggests increased cloud life time, which is contrary to this mechanism.

Contrary to the first and second aerosol indirect effects, the semi-direct effect cannot be excluded as an explanatory process, since the signs of changes of all aerosol and cloud variables presented here are consistent with what would be expected based on this mechanism. Specifically, this effect predicts that decreasing absorbing aerosols inside the cloud layers would lead to reduced evaporation of cloud droplets and hence increased cloudiness and cloud water content. It is important noting that this mechanism holds primarily for absorbing aerosols, such as biomass burning particles, which is the case in this study. It

is also important noting that the position of the aerosols relative to the cloud layer determines the sign of the semi-direct effect: a decrease in aerosols will lead to increased cloudiness only if the aerosols are at the same level with clouds. If the aerosols are above clouds, the effect will be the opposite (Koch and Del Genio, 2010).

### 3.3.3 Profiles of aerosol and cloud changes

In order to further examine the possibility of the semi-direct effect as an underlying mechanism, an analysis of the vertically

resolved changes in aerosol extinction profiles was conducted, based on CALIPSO data, combined with typical values of cloud extinction profiles for this region. October and November were selected as the most characteristic months in terms of aerosol changes, since both exhibit a significant decrease in aerosols, which however is attributed to different aerosol types, namely smoke aerosols for October and polluted dust for November (Fig. 3b). Figure 8a shows the typical profile of cloud extinction in autumn over South China, available from the LIVAS data set (Lidar climatology of Vertical Aerosol Structure

for space-based lidar simulation studies; Amiridis et al., 2015) based on measurements from 2007 to 2011. It is apparent that low clouds prevail during this season. Figures 8b and 8c show, for the same height range, changes in the aerosol extinction profiles in October and November during 2006-2015, separately for total, dust, smoke and polluted dust aerosols. In October, changes in smoke prevail, in agreement with Fig. 3b. When compared with the cloud extinction profile, it appears that the decrease in smoke aerosols tended to occur mostly above clouds. In November, however, polluted dust is the leading

decreasing aerosol type (see also Fig. 3b) and this decrease was more pronounced towards the surface. In fact, the shape of the profile change suggests that most of the November decrease occurred near or within clouds. It should be noted here, that the uncertainty in aerosol extinction profiles retrieval from CALIPSO increases in lower atmospheric layers (Young et al., 2013), thus decreasing the confidence in the results towards the surface. The vertically resolved analysis of aerosol type changes showed that the significance level for smoke in October (Fig. 8b) lies between 90% and 95% in the range between 1

km and 1.5 km altitude, while polluted dust changes in November are significant in the 95% confidence interval between 0.7 km and 1.2 km.

Analysis of MODIS level 3 data for these same months, classified according to the International Satellite Cloud Climatology Project (ISCCP) cloud type classification scheme, shows that stratocumulus clouds prevail over the region during this season. While this is shown in Fig. 9a on an autumn average basis, the pattern is similar if October and November are

examined separately. Stratocumulus clouds also exhibit the largest changes, with a considerable decrease in October (Fig. 9b) and increase in November (Fig. 9c), although not in a statistically significant sense. Nevertheless, these results show consistency with an aerosol semi-direct effect mechanism acting under decreasing aerosol loads. Specifically, the decrease in



biomass burning aerosols within clouds (November case) coincides with an increase in liquid cloud fraction and water content (Figs. 7a, 7b), attributed primarily to stratocumulus clouds (Fig. 9c), while the decrease in smoke aerosols above clouds (October case) coincides with a considerable decrease in stratocumulus CFC (Fig. 9b). In both these months, the positions of aerosols and clouds and their signs of changes agree well with the different semi-direct effect mechanisms

reported for different aerosol-cloud configurations: less absorbing aerosols above stratocumulus clouds would lead to a decrease in CFC due to a weakened inversion and enhanced cloud-top entrainment (as in October), while less absorbing aerosols within clouds would lead to more and thicker clouds, by reducing cloud evaporation (as in November).

The results on aerosol profile changes also agree with our previous conclusions on the origins of aerosols, based on Figs. 3b and 3c. A decrease in smoke aerosols that are transported from remote areas (as in the October case) would probably occur at

higher altitudes, whereas a decrease in aerosols from local sources (as in polluted dust in November) is expected to be proportional to their typical profile (higher concentrations at lower atmospheric levels). Further analysis of the monthly time series showed that the liquid CFC and the polluted dust AOD are anti-correlated in November and December, with correlation coefficients around -0.7 to -0.8. This anti-correlation is not apparent in other months, and the decreasing pattern in the polluted dust profile in November is not present for the other aerosol types or months.

**4 Summary**

In the present study, aerosol and cloud characteristics and changes were analysed based on a synergistic use of multiple independent remote sensing data sets. The study focused on the South China region, which is characterised by intense aerosol-producing human activities, while a significant decrease in aerosol loads has previously been reported. In agreement to these previous reports, it was found that absorbing aerosol loads over the region decreased significantly, and this decrease

was attributed mainly to changes in biomass burning activities. Concurrent changes in liquid cloud fraction and thickness were observed, with notable increases and decreases in different months. Further analysis of vertical profiles of both aerosols and clouds showed that the signs of cloud changes depended on the position of aerosols relative to clouds, being in agreement with the predictions of the aerosol semi-direct effect, under different aerosol and cloud configurations.

The aerosol semi-direct effect has been studied in the past through both model simulations (e.g. Allen and Sherwood, 2010;

Ghan et al., 2012) and analysis of observations (e.g. Wilcox, 2012; Amiri-Farahani et al., 2017). While its magnitude on a global average scale appears less pronounced compared to indirect aerosol effects, it has been shown that on local scales and in specific aerosol-cloud regimes its consequences can be significant. Here, the combined analysis of different aerosol and cloud data sets showed a high level of consistency with predictions of this mechanism. It should be stressed however, that apart from strong indications, these results do not constitute evidence of any cause and effect mechanism, which cannot be

proved based on observations only. They rather represent a contribution to the observational approaches in aerosol-cloud-radiation interaction studies, highlighting both the possibilities and limitations of these approaches. To overcome some of these limitations, further research will focus on model simulations of the conditions described here, in order to provide more insights regarding the underlying physical mechanism.

**Acknowledgments**

We acknowledge the EUMETSAT CM SAF project for providing support to this study. MODIS aerosol and cloud data were obtained from https://ladsweb.modaps.eosdis.nasa.gov; CALIPSO aerosol data were obtained from https://eosweb.larc.nasa.gov/project/calipso/cal_lid_l3_apro_allsky-standard-v3-00; GFED data were obtained from http://www.geo.vu.nl/~gwerf/GFED/GFED4; CLARA-A2 cloud data were obtained from http://www.cmsaf.eu; LIVAS cloud data were obtained from http://lidar.space.noa.gr:8080/livas.



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



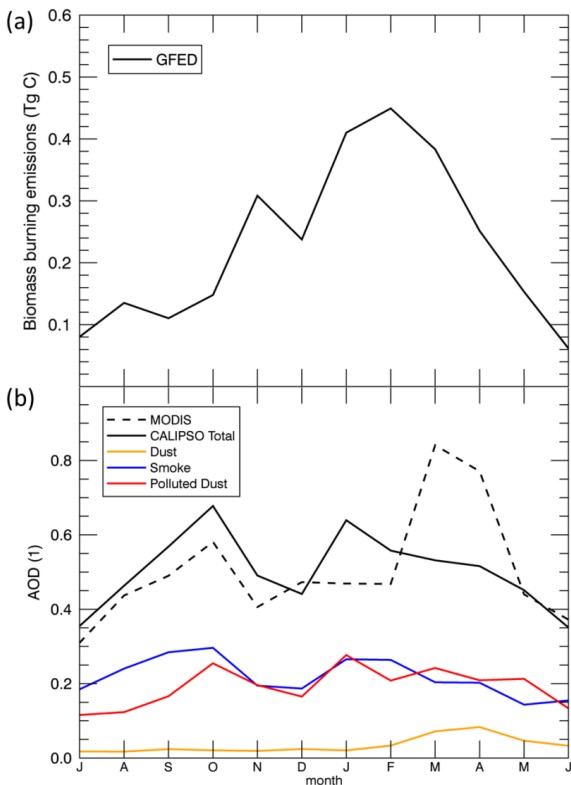

**Figure 1. Seasonal variations in emissions and aerosols over South China, based on the period 2006-2015. (a) GFED biomass burning emissions (Tg C), (b) AOD from MODIS and CALIPSO, including CALIPSO components of dust, smoke and polluted dust AOD. Note that the horizontal axis starts in June and ends in July.**




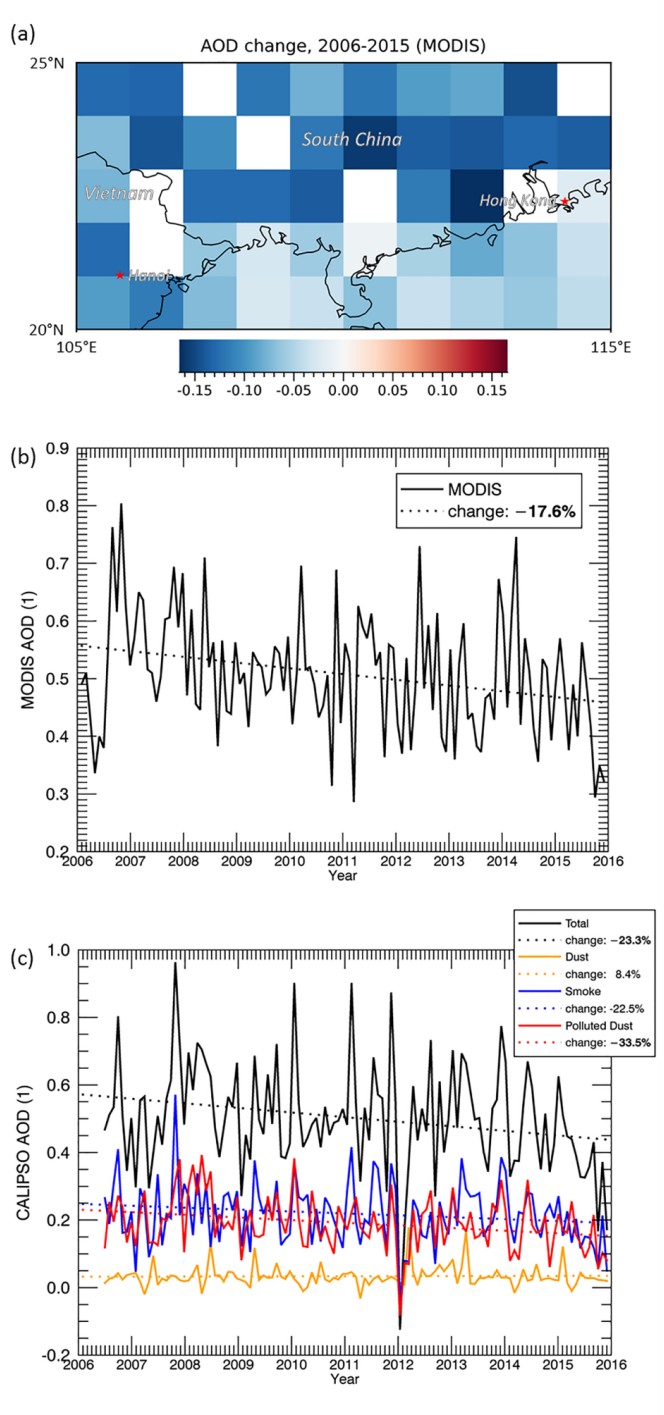

**Figure 2. Changes in AOD over South China during 2006-2015. (a) Spatial distribution of AOD change over the study region deduced from MODIS data. Spatially averaged monthly deseasonalized values of AOD from MODIS (b) and CALIPSO, including components of dust, smoke and polluted dust AOD (c). Dotted lines correspond to linear regression fits. Percent changes during**
5 **the period examined are also shown, with the statistically significant ones indicated in bold.**




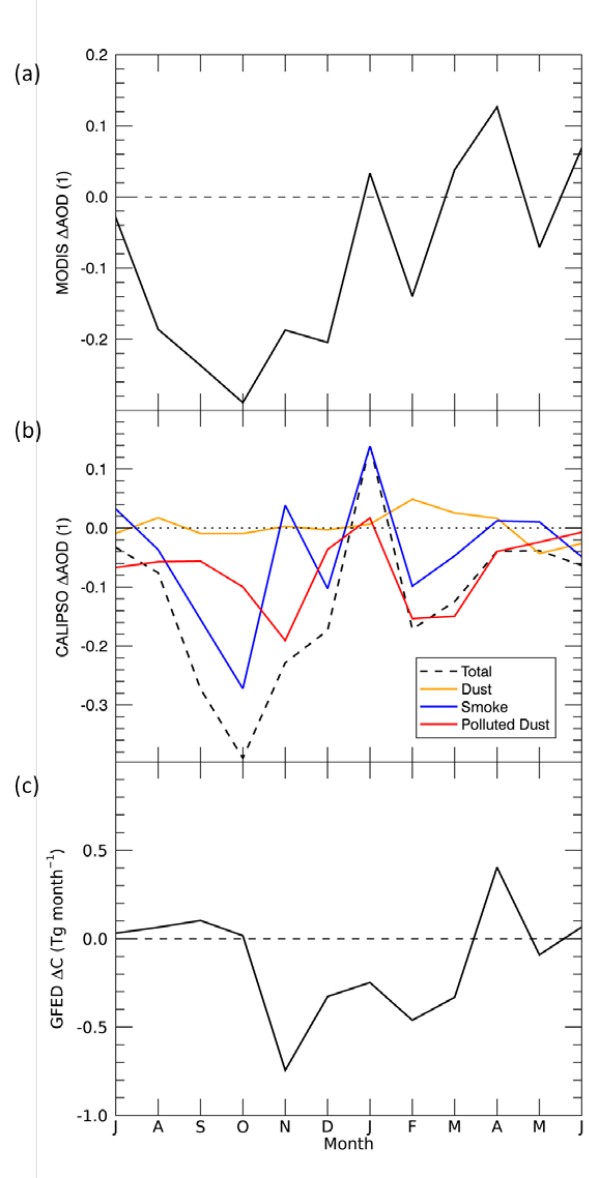

**Figure 3. Seasonal variation of changes in aerosols and emissions over South China. (a) AOD changes from 2006 to 2015 deduced from MODIS data. (b) AOD changes from 2007 to 2015 deduced from CALIPSO data, in total and per aerosol type. (c) Biomass**
5 **burning aerosol emission changes from 2006 to 2015 based on GFED data.**



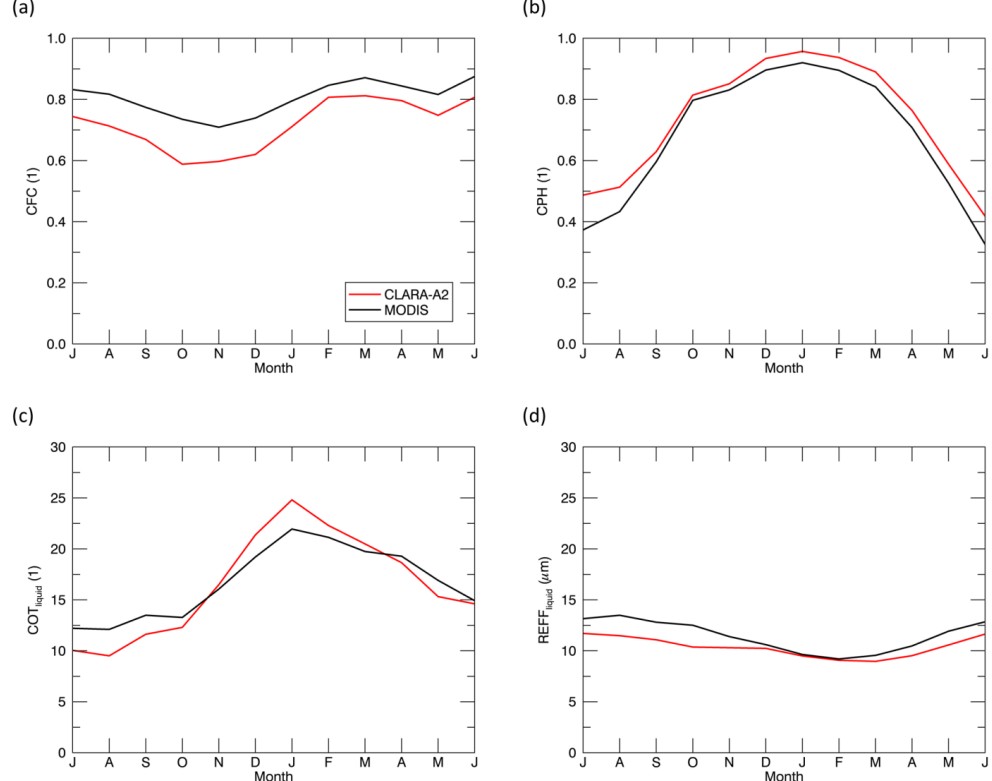

**Figure 4. Seasonal variations in cloud properties over South China, based on CLARA-A2 and MODIS data, during the period 2006-2015. (a) Total CFC, (b) cloud phase (CPH; fraction of liquid clouds relative to total CFC), (c) COT for liquid clouds and (d) REFF.**





**Figure 5. Changes in cloud properties over South China during 2006-2015, based on CLARA-A2 and MODIS data. (a), (b) Spatial distributions of changes in all-sky LWP and liquid CFC based on CLARA-A2 data. Spatially averaged monthly deseasonalized values of all-sky LWP (c), liquid CFC (d), liquid COT (e) and REFF (f).**




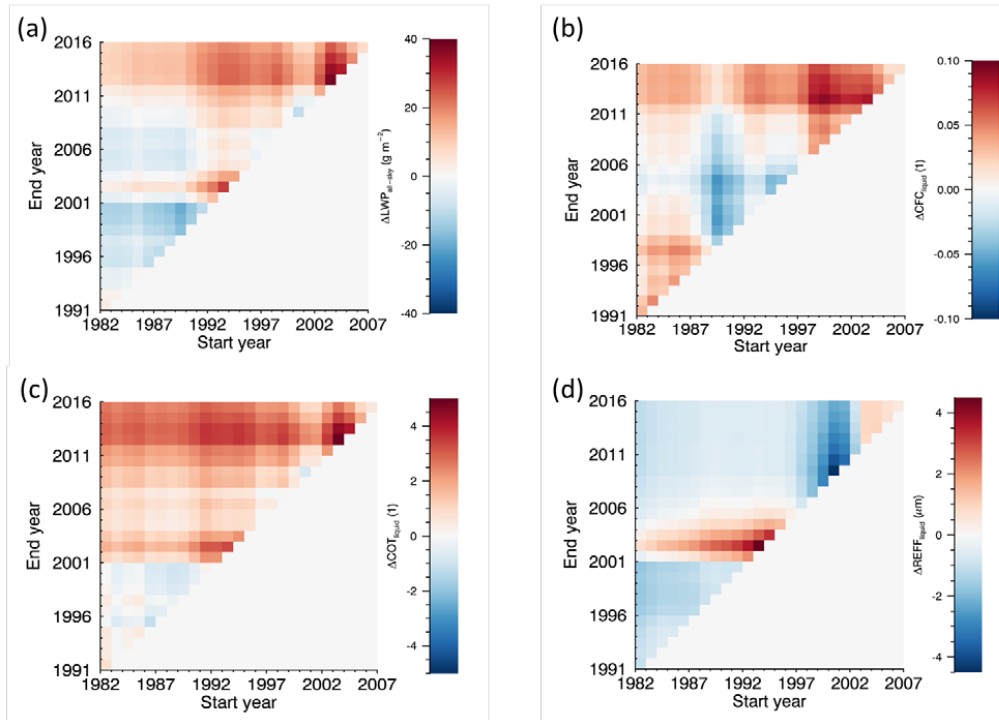

**Figure 6.** Changes in liquid cloud properties over South China, based on 34 years of CLARA-A2 data (1982-2015) and estimated for all possible combinations of start and end years, with a minimum time range of 10 years. The four plots show corresponding changes in (a) all-sky LWP, (b) liquid CFC, (c) liquid COT and (d) liquid REFF.





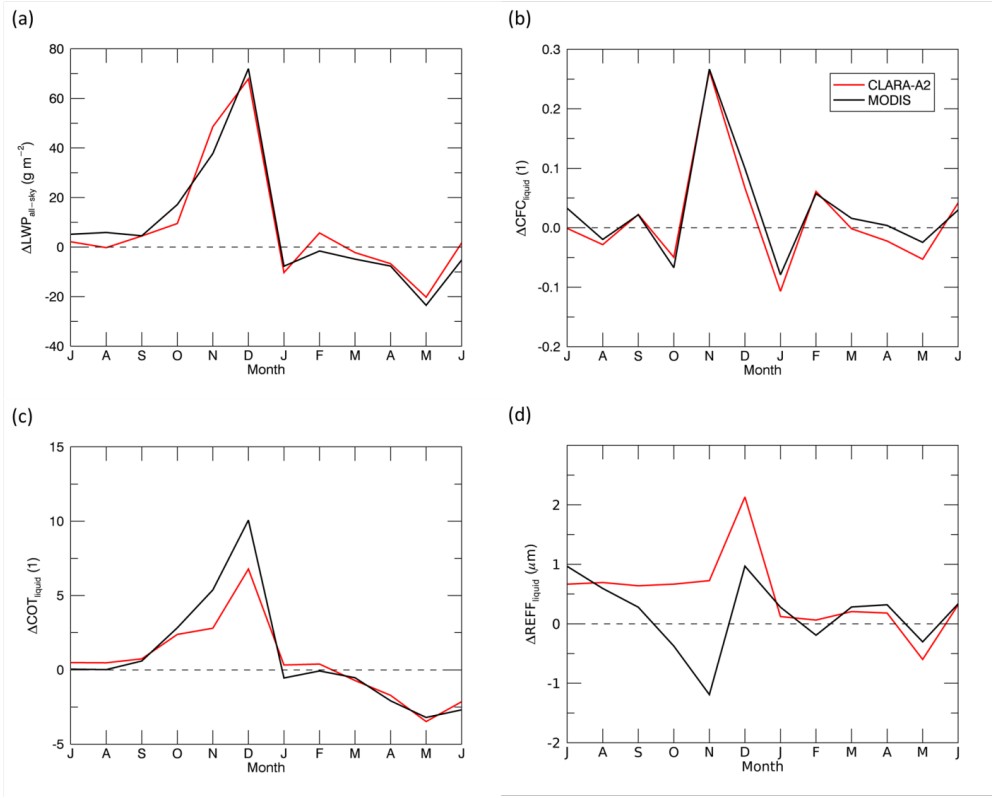

**Figure 7. Seasonal variation of changes in liquid cloud properties over South China. (a) all-sky LWP, (b) liquid CFC, (c) liquid COT and (d) liquid REFF changes from 2006 to 2015 based on CLARA-A2 and MODIS data.**



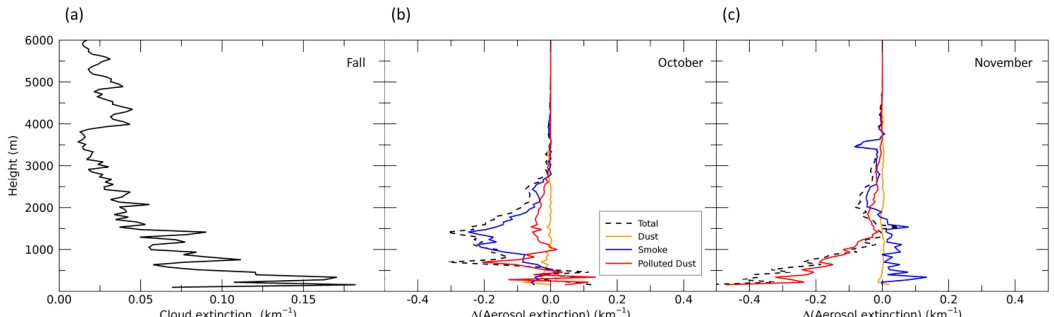

**Figure 8. Profiles of cloud and aerosol changes over South China. (a) Cloud extinction in autumn (September-November), estimated based on LIVAS CALIPSO data from 2007-2011. Aerosol extinction change for October (b) and November (c) based on CALIPSO level 3 data from 2007-2015. Changes are estimated for total aerosol and separately for dust, smoke and polluted dust aerosol types.**





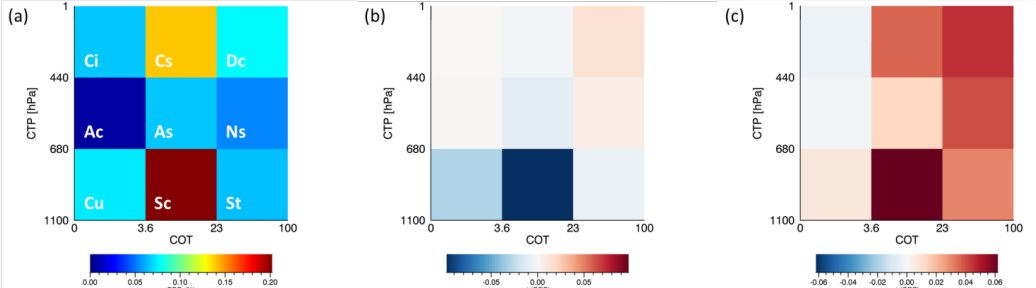

**Figure 9. (a) Average autumn CFC for different cloud types, from MODIS data and the ISCCP cloud classification scheme based on joint COT – CTP (cloud top pressure) histograms. Cloud type abbreviations are as follows: Ci: Cirrus, Cs: Cirrostratus, Dc: Deep convection, Ac: Altocumulus, As: Altostratus, Ns: Nimbostratus, Cu: Cumulus, Sc: Stratocumulus, St: Stratus. Changes in**

5 **CFC for each cloud type in October and November are shown in (b) and (c), respectively.**