# Peer review of "Satellite observations of aerosols and clouds over South China from 2006 to 2015: analysis of changes and possible interactions"

_Atmospheric Chemistry and Physics, 2018_

## Referee Comment (RC1) · Anonymous Referee #1 · 2 Aug 2018

This study used multi-source data to investigate the aerosol and cloud properties over South China and discussed the potential mechanism. This work is meaningful; however, it lacks innovation in technical regard and some conclusions/discussions are incorrect. From the satellite remote sensing, the authors may misunderstand the definition of CALIPSO aerosol type (see major comments). I also find some flaws/errors. The specific comments are as following:

Major comments: The author listed three CALIPSO aerosol types in section 3.1: smoke, polluted dust, and dust. Aerosol type is related to the optical properties of aerosol. Although "polluted dust" is the mixture of dust and smoke, "polluted dust" is

a type of aerosol, not the simple integration of "smoke" and "dust". However, it seems that authors regarded "polluted dust" as the simple integration of dust and smoke, and on this basis, they made conclusions/discussions in Section 3.1, e.g., deduced "the decrease in polluted dust AOD can also be attributed to biomass burning aerosols" and "the changes in polluted dust should also be attributed to reductions in biomass burning aerosols". For example, if dust and smoke did not change, but total AOD and polluted dust decreased. In this case, how to explain?

The other specific comments: 1. Page 3 line 4-10: please point out the scientific dataset's name for cloud data.

2. Page 3 line 22-25: please rewrite this sentence. "initial and final" may cause misunderstanding.

3. Page 3 line 24-26: why not use slope of the regression line to examine the change? 4. Page 3 line 35: rewrite this sentence. Biomass burning is not the only aerosol source in South China.

5. Page 4 line 8-10: please explain why the differences in March and April reached the maximum?

6. Page 4 line 28: not recommend cite an ACPD paper. A paper which is under review may have errors. There are other papers, like Ma Z. et al (2016) and He Q. et al (2016).

7. Page 4 line 40-44: what is "C" emissions? It means GFED? Why GFED partially agrees with the total AOD change pattern can infer aerosols over the study area is transported from neighboring region? There is no any other type of emission? All aerosols in South China come from biomass burning?

8. Figure 1: the horizontal axis may start in July and ends in June.

---

## Referee Comment (RC2) · Anonymous Referee #2 · 7 Aug 2018

The authors investigated seasonal and decadal variations of aerosols and clouds over South China using several satellite observation data and GFED biomass burning emissions to understand aerosol cloud interactions and aerosol semi-direct effect. The methodology of combined use of passive and active satellite sensors is useful, but some discussions are not enough clear to understand aerosols cloud interactions.

General Comments:

1. The authors used the CALIPSO aerosol classification product to examine atmospheric aerosol composition over South China; however, misclassification of aerosol type in CALIPSO product often occurs. Burton et al. (2013) indicated that 78% of

the smoke layers of the CALIPSO product are inferred by the airborne high spectral resolution lidar (HSRL) to be urban (polluted continental) aerosol.

2. The accuracy of data products and the uncertainties caused by the different data sampling derived from the different sensors are not discussed in the manuscript. More detailed description about data quality is needed.

Specific comments:

P2 line 25: The paper about the CALIPSO level 3 product written by Tackett et al. (2018) was recently published. I would suggest that the authors cite this paper.

P2 line 28: Why only three types (dust, smoke, and polluted dust) were used? The CALIPSO aerosol models are consist of six aerosol types (Omar et al. 2009).

P2 line 33: Why monthly GFED data used? I think the daily product is more appropriate for applying the thresholds in section 2.3.

P3 line 26: It is not clear to me how several products with different pixel sizes are treated. In addition, how about data sampling? The data sampling of each sensor is different. The swath width of the MODIS sensor is 2330 km, while the CALIPSO lidar only measure the nadir direction from the satellite orbit.

P4 line 9: It is not clear to me why MODIS AOD and CALIPSO AOD are different in Mach and April. The difference of MODIS AOD and CALIPSO total AOD in March and April is 0.3, which is comparable to the seasonal variation of CALIPSO total AOD.

P7 line 23: The CALIPSO lidar is unable to detect aerosols and clouds underneath optically dense cloud layers; therefore, the extinction coefficient of low-level cloud in Figure 8a is underestimated.

P8 line 5: "less absorbing aerosols above stratocumulus clouds would lead . . .". The "polluted continental" type of CALIPSO aerosol models (Omar et al., 2009) is also regarded as one of absorbing aerosols. Why the "polluted continental" type is not

included in the analysis?

References:

Burton, S. P., Ferrare, R. A., Vaughan, M. A., Omar, A. H., Rogers, R. R., Hostetler, C. A., and Hair, J. W.: Aerosol classification from airborne HSRL and comparisons with the CALIPSO vertical feature mask, Atmos. Meas. Tech., 6, 1397-1412, https://doi.org/10.5194/amt-6-1397-2013, 2013.

Tackett, J. L., Winker, D. M., Getzewich, B. J., Vaughan, M. A., Young, S. A., and Kar, J.: CALIPSO lidar level 3 aerosol profile product: version 3 algorithm design, Atmos. Meas. Tech., 11, 4129-4152, https://doi.org/10.5194/amt-11-4129-2018, 2018.

---

## Author Comment (AC1) · 7 Aug 2018

**First reply to RC1 comment of Anonymous Referee #1 on the ACPD paper "Satellite observations of aerosols and clouds over South China from 2006 to 2015: analysis of changes and possible interactions" by Nikos Benas et al. (Referee comments are repeated in italics).**

**General statement:**

*This study used multi-source data to investigate the aerosol and cloud properties over South China and discussed the potential mechanism. This work is meaningful; however, it lacks innovation in technical regard and some conclusions/discussions are incorrect. From the satellite remote sensing, the authors may misunderstand the definition of CALIPSO aerosol type (see major comments). I also find some flaws/errors. The specific comments are as following:*

**Reply:**

We thank the Referee for reviewing our manuscript. As we tried to explain in the paper (e.g., page 2, lines 8-13), we consider that its innovativeness lies more in the combined usage of different satellite-based data sets, and the examination of their potential and limitations for explaining the observed changes in aerosol and cloud properties, rather than in technical aspects, since rather standard methods were adopted. We plan to emphasize this innovative aspect further in the abstract of the revised manuscript. Following are our replies to the major and specific comments:

**Major comment:**

*Major comments: The author listed three CALIPSO aerosol types in section 3.1: smoke, polluted dust, and dust. Aerosol type is related to the optical properties of aerosol. Although "polluted dust" is the mixture of dust and smoke, "polluted dust" is a type of aerosol, not the simple integration of "smoke" and "dust". However, it seems that authors regarded "polluted dust" as the simple integration of dust and smoke, and on this basis, they made conclusions/discussions in Section 3.1, e.g., deduced "the decrease in polluted dust AOD can also be attributed to biomass burning aerosols" and "the changes in polluted dust should also be attributed to reductions in biomass burning aerosols". For example, if dust and smoke did not change, but total AOD and polluted dust decreased. In this case, how to explain?*

**Reply:**

Based on Omar et al. (2009), who describe the CALIPSO aerosol classification algorithm, "polluted dust" is an aerosol type considered separately from the "smoke" and "dust" types, as the Referee also explains. In that study it is stated that the polluted dust model "is designed to account for episodes of dust mixed with biomass burning smoke", and also "for instances of dust mixed with urban pollution". In our study we also analyze the "polluted dust" AOD as an independent aerosol source, which may contain aerosols from biomass burning. In this context, we agree that the statement in page 4, lines 25-26 ("Since no significant change is found for pure dust aerosols, the decrease in polluted dust AOD can also be attributed to biomass burning aerosols"), repeated in page 4, lines 33-34, can be misleading, since it implies a direct connection between "dust" and "polluted dust" aerosols. Although a decrease in biomass burning aerosols is a possible explanation for the decrease in polluted dust AOD, corroborated by the similarity in the corresponding changes in GFED emissions (Figs. 3b and 3c), it is not the only one. This will be clearly stated in the revised manuscript. In our opinion, however, this point does not affect significantly the relevant conclusions and discussions. Specifically, the large decrease in polluted dust AOD in November (Figure 3b),

combined with the corresponding shape of the AOD profile change (Figure 8c), and the fact that the polluted dust model is a combination of a desert dust and a biomass burning mode, as explained in Omar et al. (2009), still point to a concurrence of changes in an absorbing aerosol type and low (mainly Stratocumulus) clouds (based on Figure 9c). This concurrence led to the relevant discussion on possible explanatory mechanisms (Section 3.3.3).

**Specific comment 1:**
*Page 3 line 4-10: please point out the scientific dataset's name for cloud data.*
**Reply:**
In the revised manuscript we will include a table containing the scientific dataset names of all cloud variables used from MODIS and CLARA-A2.

**Specific comment 2:**
*Page 3 line 22-25: please rewrite this sentence. "initial and final" may cause misunderstanding.*
**Reply:**
The sentence will be rewritten to avoid misunderstanding.

**Specific comment 3:**
*Page 3 line 24-26: why not use slope of the regression line to examine the change?*
**Reply:**
The methodology suggested by the Referee is analogous to the one described in the manuscript. We opted for the latter in order to provide our results in terms of absolute or percent changes during the exact period examined. Using the slope of the regression line would be more useful if changes were presented on a per year or per decade basis. The reason for selecting this method will be added in the revised manuscript.

**Specific comment 4:**
*Page 3 line 35: rewrite this sentence. Biomass burning is not the only aerosol source in South China.*
**Reply:**
This sentence was phrased using the term "include" to indicate the Referee's point: that biomass burning is not the only aerosol source in South China. We will rephrase it in the revised manuscript, in order to emphasize this point further.

**Specific comment 5:**
*Page 4 line 8-10: please explain why the differences in March and April reached the maximum?*
**Reply:**
An explanation of this point is not straightforward. Based on our analysis, there is no apparent characteristic of either aerosols or clouds appearing in these two particular months, that could help explaining these differences between MODIS and CALIPSO. Hence, further investigation is required for providing a definitive explanation of these differences, which would be beyond the scope of the present study. Instead, we provide a general description of known reasons that cause differences between MODIS and CALIPSO AOD.

**Specific comment 6:**

*Page 4 line 28: not recommend cite an ACPD paper. A paper which is under review may have errors. There are other papers, like Ma Z. et al (2016) and He Q. et al (2016).*

**Reply:**

We thank the Referee for this recommendation. The suggested papers will be included in the revised manuscript.

**Specific comment 7:**

*Page 4 line 40-44: what is "C" emissions? It means GFED? Why GFED partially agrees with the total AOD change pattern can infer aerosols over the study area is transported from neighboring region? There is no any other type of emission? All aerosols in South China come from biomass burning?*

**Reply:**

"C" is the total mass of carbon particles (given in Tg), available from the GFED data set. The discrepancies in changes between biomass burning emissions and satellite-derived AOD could indeed be caused by aerosols transported from other regions, or by other types of emissions not included in GFED (e.g. sulfates), as the Referee suggests and we also mention in page 4, lines 14-15, or by both. Our analysis, however, does not imply that biomass burning is the only type of aerosol emissions in South China. In fact, as we describe in other parts of the manuscript, based on findings from recent studies (page 3, lines 35-39 and page 4, lines 1-2), biomass burning constitutes one of the main aerosol sources over the area, originating from different activities with different seasonal characteristics, with domestic burning being the main contributor in autumn and winter. Our point here is that if a change in aerosols from biomass burning (smoke aerosols in our case) is not well captured by the biomass burning emissions data set, it is likely that this change occurred in a nearby region. We plan to rephrase these lines in the revised manuscript, trying to clarify this point.

**Specific comment 8:**

*Figure 1: the horizontal axis may start in July and ends in June.*

**Reply:**

We thank the Referee for this point. The Figure 1 caption will be corrected accordingly.

---

## Author Comment (AC2) · 17 Aug 2018

**First reply to RC2 comment of Anonymous Referee #2 on the ACPD paper "Satellite observations of aerosols and clouds over South China from 2006 to 2015: analysis of changes and possible interactions" by Nikos Benas et al. (Referee comments are repeated in italics).**

**General statement:**

*The authors investigated seasonal and decadal variations of aerosols and clouds over South China using several satellite observation data and GFED biomass burning emissions to understand aerosol cloud interactions and aerosol semi-direct effect. The methodology of combined use of passive and active satellite sensors is useful, but some discussions are not enough clear to understand aerosols cloud interactions.*

**Reply:**

We thank the Referee for reviewing our manuscript. Following are our replies to the general and specific comments:

**General comments:**

*1. The authors used the CALIPSO aerosol classification product to examine atmospheric aerosol composition over South China; however, misclassification of aerosol type in CALIPSO product often occurs. Burton et al. (2013) indicated that 78% of the smoke layers of the CALIPSO product are inferred by the airborne high spectral resolution lidar (HSRL) to be urban (polluted continental) aerosol.*

**Reply:**

The primary purpose of using CALIPSO aerosol data in our analysis was the inclusion of information on the vertical distribution of aerosols, which is crucial in assessing the possibility of aerosol-cloud interactions. The importance of this information is highlighted in Section 3.3.3 specifically for the different semi-direct effect manifestations, which occur with absorbing aerosols present. Regarding the sources of these aerosol types, we infer some possibilities based on comparisons of their changes with corresponding changes in GFED emissions (page 4, lines 39-43 and page 5, lines 1-3). These conclusions, however, are not definitive, and we plan to rephrase these parts accordingly. The comment of the Referee also points to the same ambiguities in these conclusions, since the mentioned misclassification suggests differences in aerosol origins and compositions. It is a crucial addition to the relevant discussion and will be included in the revised manuscript. It should also be noted, however, that the misclassifications of smoke and polluted dust aerosols reported in Burton et al. (2013) most of the times replace an absorbing aerosol type with another absorbing type. While this is crucial when trying to pinpoint the origin of these aerosols, it would not affect significantly the discussion in Section 3.3.3, on how an absorbing aerosol layer would interact with clouds based on their relative position.

*2. The accuracy of data products and the uncertainties caused by the different data sampling derived from the different sensors are not discussed in the manuscript. More detailed description about data quality is needed.*

**Reply:**

In order to determine the accuracy of the data products used, validation against reference measurements is required. While this would be beyond the scope of this study, we agree that a relevant discussion is missing, and we plan to refer to relevant studies in the revised manuscript. Uncertainties caused by the different data sampling from the different sensors can also be an important issue. The level of agreement in cloud properties from MODIS and CLARA-A2 (Figs. 4, 5 and 7) suggests that the respective sensors, overpass times, and retrieval algorithms are indeed similar enough that possible discrepancies due to different sampling are small. This is not the case, however, when comparing CALIPSO with passive sensors, as we briefly mention in page 4, lines 8-10. For this reason we tried to minimize possible differences by ensuring that all data used are representative of the same area, months and time periods examined. This was performed by applying the common threshold criteria described in Section 2.3. We intend to emphasize this point in the revised manuscript. We will also include in Section 2 a more detailed description of the quality criteria adopted for the creation of the level 3 data used here, and possible effects of these in our findings.

**Specific comments:**

*P2 line 25: The paper about the CALIPSO level 3 product written by Tackett et al. (2018) was recently published. I would suggest that the authors cite this paper.*
**Reply:**
We thank the Referee for this suggestion. A citation will be included in the revised manuscript.

*P2 line 28: Why only three types (dust, smoke, and polluted dust) were used? The CALIPSO aerosol models are consist of six aerosol types (Omar et al. 2009).*
**Reply:**
As explained in our reply to the next comment, we used monthly averaged data products. The corresponding aerosol profile product from CALIPSO is the level 3, version 3 monthly product. It is interesting that, while the CALIPSO aerosol models consist indeed of six types, only three of them are available in level 3 (dust, smoke, and polluted dust). The reason is given in the discussion of the relevant paper (Tackett et al., 2018), where the authors mention that they chose to include a subset of aerosol subtypes for better management of the level 3 file sizes. According to their reply, the selection of these three subtypes was decided based on the algorithm performance in detecting them (Atmos. Meas. Tech. Discuss., doi: 10.5194/amt-2018-97-AC2, 2018). This is an important piece of information regarding this data product, that we plan to include in the revised manuscript. We acknowledge that a similar analysis of all aerosol types would probably provide more robust conclusions. It would require, however, a level of data processing similar to the one used to create the CALIPSO level 3 data set in the first place. This would exceed the scope of the present study. Furthermore, the additional analysis would not alter significantly the conclusions regarding the three aerosol types already analyzed.

*P2 line 33: Why monthly GFED data used? I think the daily product is more appropriate for applying the thresholds in section 2.3.*
**Reply:**
All data sets used in this study were monthly averages. We consider this temporal scale appropriate for studying both long-term interannual (many years or decades) as well as seasonal changes.

However, the application of the first threshold (page 3, lines 26-27) requires the number of days used in the calculation of the monthly average. This information is available in all level 3 data sets used. In GFED, the non-zero "daily_fraction" values, available on a daily basis and attributing a daily fraction to the total monthly emissions, were used as a count of the days used. It should be noted here, that monthly values of emissions over the study area were calculated by summing the corresponding pixel values, instead of averaging, which was the case for all other variables. Accordingly, in the case of GFED, this first threshold was applied on a "whole area", rather than a pixel basis. We plan to include these details in the revised manuscript.

*P3 line 26: It is not clear to me how several products with different pixel sizes are treated. In addition, how about data sampling? The data sampling of each sensor is different. The swath width of the MODIS sensor is 2330 km, while the CALIPSO lidar only measure the nadir direction from the satellite orbit.*
**Reply:**
Our analysis and corresponding results were based on spatially and temporally averaged values of cloud and aerosol properties, rather than pixel-level and instantaneous comparisons. In this context, we tried to ensure a fair treatment among data sets with different pixel sizes and samplings by applying common threshold criteria to the monthly and spatially averaged values, as described in Section 2.3. We also tried to minimize differences due to temporal sampling by only using data from afternoon satellites: MODIS-Aqua, AVHRRs on NOAA-18 and -19 (for CLARA-A2) and the corresponding daytime product from CALIPSO. It is true, however, that the viewing geometry of CALIPSO and the passive sensors is substantially different and can cause large discrepancies. While the underlying assumption here is that monthly averages smooth out these differences, we also acknowledge this possibility in page 4, lines 8-10. We consider it appropriate, however, to emphasize it further in Section 2.3 of the revised manuscript.

*P4 line 9: It is not clear to me why MODIS AOD and CALIPSO AOD are different in Mach and April. The difference of MODIS AOD and CALIPSO total AOD in March and April is 0.3, which is comparable to the seasonal variation of CALIPSO total AOD.*
**Reply:**
The difference in total AOD between the two data sets in March and April is indeed remarkable. However, we could not identify possible reasons for these discrepancies based on the data sets analyzed in this study. While this feature certainly merits further investigation, we decided not to provide hypotheses that cannot be supported by the present analysis. Since a possible explanation requires analyzing data from additional sources, we consider it as a significant digression from the present analysis, also considering that the main conclusions of this study were not drawn from features appearing in these two months. We plan, however, to highlight the need for further investigation in the revised manuscript.

*P7 line 23: The CALIPSO lidar is unable to detect aerosols and clouds underneath optically dense cloud layers; therefore, the extinction coefficient of low-level cloud in Figure 8a is underestimated.*
**Reply:**
We thank the Referee for this clarification, which supports our conclusion that low clouds prevail over the area during this season. We plan to include it in the revised manuscript.

*P8 line 5: "less absorbing aerosols above stratocumulus clouds would lead : : :". The "polluted continental" type of CALIPSO aerosol models (Omar et al., 2009) is also regarded as one of absorbing aerosols. Why the "polluted continental" type is not included in the analysis?*

**Reply:**

Please refer to our reply in the second specific comment.